# ParaQA: A Question Answering Dataset with Paraphrase Responses for Single-Turn Conversation

Endri Kacupaj[1], Barshana Banerjee[1], Kuldeep Singh[2], and Jens Lehmann[1,3]

[1] University of Bonn, Bonn, Germany
{kacupaj,jens.lehmann}@cs.uni-bonn.de s6babane@uni-bonn.de
[2] Zerotha Research and Cerence GmbH, Germany
kuldeep.singh1@cerence.com
[3] Fraunhofer IAIS, Dresden, Germany,
jens.lehmann@iais.fraunhofer.de

**Abstract.** This paper presents ParaQA, a question answering (QA) dataset with multiple paraphrased responses for single-turn conversation over knowledge graphs (KG). The dataset was created using a semi-automated framework for generating diverse paraphrasing of the answers using techniques such as back-translation. The existing datasets for conversational question answering over KGs (single-turn/multi-turn) focus on question paraphrasing and provide only up to one answer verbalization. However, ParaQA contains 5000 question-answer pairs with a minimum of two and a maximum of eight unique paraphrased responses for each question. We complement the dataset with baseline models and illustrate the advantage of having multiple paraphrased answers through commonly used metrics such as BLEU and METEOR. The ParaQA dataset is publicly available on a persistent URI for broader usage and adaptation in the research community.

**Keywords:** Question Answering · Paraphrase Responses · Single-Turn Conversation · Knowledge Graph · Dataset.

**Resource Type:** Dataset
**License:** Attribution 4.0 International (CC BY 4.0)
**Permanent URL:** https://figshare.com/projects/ParaQA/94010

## 1 Introduction

In recent years, publicly available knowledge graphs (e.g., DBpedia [21], Wikidata [40]) and Yago [36]) have been widely used as a source of knowledge in several tasks such as entity linking, relation extraction, and question answering [22]. Question answering (QA) over knowledge graphs, in particular, is an essential task that maps a user's utterance to a query over a knowledge graph (KG) to retrieve the correct answer [34]. With the increasing popularity of intelligent personal assistants (e.g., Alexa, Siri), the research focus has been shifted to conversational question answering over KGs that involves single-turn/multi-turn

Table 1: Comparison of ParaQA with existing QA datasets over various dimensions. Lack of paraphrased utterances of answers remains a key gap in literature.

| Dataset | Large scale(>=5K) | Complex Questions | SPARQL | Verbalized Answer | Paraphrased Answer |
|---|:---:|:---:|:---:|:---:|:---:|
| ParaQA (This paper) | ✓ | ✓ | ✓ | ✓ | ✓ |
| Free917 [8] | ✗ | ✓ | ✗ | ✗ | ✗ |
| WebQuestions [4] | ✓ | ✗ | ✗ | ✗ | ✗ |
| SimpleQuestions [6] | ✓ | ✗ | ✓ | ✗ | ✗ |
| QALD (1-9)[4] | ✗ | ✓ | ✓ | ✗ | ✗ |
| LC-QuAD 1.0 [38] | ✓ | ✓ | ✓ | ✗ | ✗ |
| LC-QuAD 2.0 [10] | ✓ | ✓ | ✓ | ✗ | ✗ |
| ComplexQuestions [3] | ✗ | ✓ | ✗ | ✗ | ✗ |
| ComQA [1] | ✓ | ✓ | ✗ | ✗ | ✗ |
| GraphQuestions [35] | ✓ | ✓ | ✓ | ✗ | ✗ |
| ComplexWebQuestions [37] | ✓ | ✓ | ✓ | ✗ | ✗ |
| VQuAnDa [20] | ✓ | ✓ | ✓ | ✓ | ✗ |
| CSQA [31] | ✓ | ✓ | ✗ | ✗ | ✗ |
| ConvQuestions [9] | ✓ | ✓ | ✗ | ✗ | ✗ |

dialogues [33]. To support wider research in knowledge graph question answering (KGQA) and conversational question answering over KGs (ConvQA), several publicly available datasets have been released [4,38,31].

**Motivation and Contributions** In dialog systems research, we can distinguish between single-turn and multi-turn conversations [7,19]. In single-turn conversations, a user provides all the required information (e.g., slots/values) at once, in one utterance. Conversely, a multi-turn conversation involves anaphora and ellipses to fetch more information from the user as an additional conversation context. The existing ConvQA [31,9] datasets provide multi-turn dialogues for question answering. In a real-world setting, user will not always require multi-turn dialogues. Therefore, single-turn conversation is a common phenomenon in voice assistants[5]. Some public datasets focus on paraphrasing the questions to provide real-world settings, such as LC-QuAD2.0 [10] and ComQA [1]. The existing ConvQA datasets provide only up to one verbalization of the response (c.f. Table 1). In both dataset categories (KGQA or ConvQA), we are not aware of any dataset providing paraphrases of the various answer utterances. For instance, given the question "How many shows does HBO have?", on a KGQA dataset (LC-QuAD [38]), we only find the entity as an answer (e.g. "38"). While on a verbalized KGQA dataset [20], the answer is verbalized as "There are 38 television shows owned by HBO." Given this context, the user can better verify that the system is indeed retrieving the total number of shows owned by HBO. However, the answer can be formulated differently using various paraphrases such as "There are 38 TV shows whose owner is HBO.", "There are 38 television programs owned by that organization" with the same semantic meaning. Hence, paraphrasing the answers can introduce more flexibility and intuitiveness in the conversations. In this paper, we argue that answer paraphrasing improves the machine learning models' performance for single-turn conversations (involving

---
[5] https://docs.microsoft.com/en-us/cortana/skills/mva31-understanding-conversations

question answering over KG) on standard empirical metrics. Therefore, we introduce ParaQA, a question answering dataset with multiple paraphrase responses for single-turn conversation over KGs.

The ParaQA dataset was built using a semi-automated framework that employs advanced paraphrasing techniques such as back-translation. The dataset contains a minimum of two and a maximum of eight unique paraphrased responses per question. To supplement the dataset, we provide several evaluation settings to measure the effectiveness of having multiple paraphrased answers. The following are key contributions of this work:

– We provide a semi-automated framework for generating multiple paraphrase responses for each question using techniques such as back-translation.
– We present ParaQA - The first single-turn conversational question answering dataset with multiple paraphrased responses. In particular, ParaQA consists of up to eight unique paraphrased responses for each dataset question that can be answered using DBpedia as underlying KG.
– We also provide evaluation baselines that serve to determine our dataset's quality and define a benchmark for future research.

The rest of the paper is structured as follows. In the next section, we describe the related work. We introduce the details of our dataset and the generation workflow in Section 3. Section 4 describes the availability of the dataset, followed by the experiments in Section 5. The reusability study and potential impact is described in Section 6. Finally, Section 7 provides conclusions.

## 2   Related Work

Our work lies at the intersection of KGQA and conversational QA datasets. We describe previous efforts and refer to different dataset construction techniques.

**KGQA Datasets** The datasets such as SimpleQuestions [6], WebQuestions [42], and the QALD challenge[6] have been inspirational for the evolution of the field. SimpleQuestions [6] dataset is one of the most commonly used large-scale benchmarks for studying single-relation factoid questions over Freebase [5]. LC-QuAD 1.0 [38] was the first large-scale dataset providing complex questions and their SPARQL queries over DBpedia. The dataset has been created using pre-defined templates and a peer-reviewed process to rectify those templates. Other datasets such as ComQA [1] and LC-QuAD 2.0 [10] are large-scale QA datasets with complex paraphrased questions without verbalized answers. It is important to note that the answers of most KGQA datasets are non-verbalized. VQuAnDa [20] is the only QA dataset with complex questions containing a single verbalized answer for each question.

---

[6] http://qald.aksw.org/

**Conversational QA**  There has been extensive research for single-turn and multi-turn conversations for open-domain [7,23,43]. The research community has recently shifted focus to provide multi-turn conversation datasets for question answering over KGs. CSQA [31] is a large-scale dataset consisting of multi-turn conversations over linked QA pairs. The dataset contained 200K dialogues with 1.6M turns and was collected through a manually intensive semi-automated process. The dataset comprises complex questions that require logical, quantitative, and comparative reasoning over a Wikidata KG. ConvQuestions [9] is a crowd-sourced benchmark with 11K distinct multi-turn conversations from five different domains ("Books", "Movies", "Soccer", "Music", and "TV Series"). While both datasets cover multi-turn conversations, none of them contains verbalized answers. Hence, there is a clear gap in the literature for the datasets focusing on single-turn conversations involving question answering over KGs. In this paper, our work combines KGQA capabilities with the conversational phenomenon. We focus on single-turn conversations to provide ParaQA with multiple paraphrased answers for more expressive conversations.

**Dataset Construction Techniques**  While some KGQA datasets are automatically generated [32], most of them are manually created either by (i) using in-house workers [38] or crowd-sourcing [10], (ii) or extract questions from online question answering platforms such as search engines, online forum, etc [4]. Most (single-turn/multi-turn) conversational QA datasets are generated using semi-automated approaches [31,9]. First, conversations are created through predefined templates. Second, the automatically generated conversations are polished by in-house workers or crowd-sourcing techniques. CSQA [31] dataset contains a series of linked QA pairs forming a coherent conversation. Further, these questions are answerable from a KG using logical, comparative, and quantitative reasoning. For generating the dataset, authors first asked pairs of in-house workers to converse with each other. One annotator in a pair acted as a user whose job was to ask questions, and the other annotator worked as the system whose job was to answer the questions or ask for clarifications if required. The annotators' results were abstracted to templates and used to instantiate more questions involving different relations, subjects, and objects. The same process was repeated for different question types such as co-references and ellipses. ConvQuestions-[9] dataset was created by posing the conversation generation task on Amazon Mechanical Turk (AMT)[7]. Each crowd worker was asked to build a conversation by asking five sequential questions starting from any seed entity of his/her choice. Humans may have an intuitive model when satisfying their real information needs via their search assistants. Crowd workers were also asked to provide paraphrases for each question. Similar to [31], the crowd workers' results were abstracted to templates and used to create more examples. While both conversational QA datasets use a relatively similar construction approach, none of them considers verbalizing the answers and providing paraphrases for them.

---

[7] https://www.mturk.com/

Table 2: Examples from ParaQA.

| Question | What is the television show whose judges is Randy Jackson? |
|---|---|
| | 1) American Idol is the television show with judge Randy Jackson. |
| Answer Verbalizations | 2) The television show whose judge Randy Jackson is American Idol. |
| | 3) The TV show he's a judge on is American Idol. |
| Question | How many shows does HBO have? |
| | 1) There are 38 television shows owned by HBO. |
| Answer Verbalizations | 2) There are 38 TV shows whose owner is HBO. |
| | 3) There are 38 television shows whose owner is that organisation. |
| | 4) There are 38 television programs owned by that organization. |
| Question | From which country is Lawrence Okoye's nationality? |
| | 1) Great Britain is the nationality of Lawrence Okoye. |
| | 2) Great Britain is Lawrence Okoye's citizenship. |
| Answer Verbalizations | 3) The nationality of Lawrence Okoye is Great Britain. |
| | 4) Lawrence Okoye is a Great British citizen. |
| | 5) Lawrence Okoye's nationality is Great Britain. |
| Question | Does Sonny Bill Williams belong in the Canterbury Bankstown Bulldogs club? |
| | 1) Yes, Canterbury-Bankstown Bulldogs is the club of Sonny Bill Williams. |
| | 2) Yes, the Canterbury-Bankstown Bulldogs is Bill Williams's club. |
| | 3) Yes, the Canterbury-Bankstown Bulldogs is his club. |
| Answer Verbalizations | 4) Yes, Canterbury-Bankstown Bulldogs is the club of the person. |
| | 5) Yes, the club of Sonny Bill Williams is Canterbury-Bankstown Bulldogs. |
| | 6) Yes, Bill Williams's club is the Canterbury-Bankstone Bulldogs. |

**Paraphrasing** In the early years, various traditional techniques have been developed to solve the paraphrase generation problem. McKeown [25] makes use of manually defined rules. Quirk et al. [30] train Statistical Machine Translation (SMT) tools on a large number of sentence pairs collected from newspapers. Wubben et al. [41] propose a phrase-based SMT model trained on aligned news headlines. Recent approaches perform neural paraphrase generation, which is often formalized as a sequence-to-sequence (Seq2Seq) learning. Prakash et al. [29] employ a stacked residual LSTM network in the Seq2Seq model to enlarge the model capacity. Hasan et al. [17] incorporate the attention mechanism to generate paraphrases. Work in [12] integrates the transformer model and recurrent neural network to learn long-range dependencies in the input sequence.

## 3 ParaQA: Question Answering with Paraphrase Responses for Single-Turn Conversation

The inspiration for generating paraphrased answers originates from the need to provide a context of the question to assure that the query was correctly understood. In that way, the user would verify that the received answer correlates with

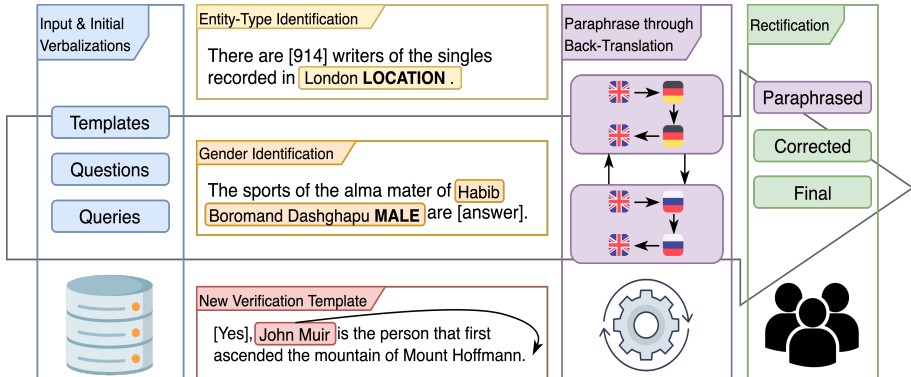

Fig. 1: Overview of dataset generation workflow. Our proposed generation workflow consists of 6 modules in total. The first module is  "**Input & Initial Verbalization**", which is responsible for producing the initial verbalized results for each input question. The next three modules ("**Entity-Type Identification though Named Entity Recognition**", "**Gender Identification**", and "**New Verification Template**") are applied simultaneously and provide new verbalized sentences based on the initial ones. Subsequently, the paraphrasing module, named "**Paraphrase through Back-Translation**", applies back translation to generated answers. Finally, in the last step ("**Rectify Verbalization**"), we rectify all the paraphrased results through a peer-review process.

the question. For illustration, in our dataset, the question "What is the commonplace of study for jack McGregor and Philip W. Pillsbury?" is translated to the corresponding SPARQL query, which retrieves the result "Yale University" from the KG. In this case, a full natural language response of the result is "Yale University is the study place of Jack McGregor and Philip W. Pillsbury.". As can be seen, this form of answer provides us the query result and details about the query's intention. At the same time, we also provide alternative paraphrased responses such as, "Yale University is the place where both Jack McGregor and Philip W. Pillsbury studied.", "Yale is where both of them studied.". All those answers clarify to the user that the question answering system completely understood the question context and the answer is correct. They can also verify that the system retrieves a place where "Jack McGregor" and "Philip W. Pillsbury" went for their studies. Table 2 illustrates examples from our dataset.

### 3.1   Generation Workflow

For generating ParaQA, we decided not to reinvent the wheel to create new questions. Hence, we inherit questions from LC-QuAD [38] and single answer verbalization of these question provided by VQuAnDa [20]. We followed a semi-automated approach to generate the dataset. The overall architecture of the approach is depicted in Figure 1.

Table 3: Examples generated from each automatic step/module of our proposed generation framework. The presented responses are the outputs from the corresponding modules before they undergo the final peer-review step. The bold text of the initial answer indicates the part of the sentence where the corresponding module is focusing. The underlined text on the generated results reveals the changes made from the module.

| | | |
|---|---|---|
| Entity-Type | Question | Count the key people of the Clinton Foundation? |
| | Initial | There are 8 key people in the **Clinton Foundation**. |
| | Generated | There are 8 key people in the organisation. |
| Gender | Question | Which planet was first discovered by Johann Gottfried Galle? |
| | Verbalized Answer | The planet **discovered by Johann Gottfried Galle** is Neptune. |
| | Generated | The planet he discovered is Neptune. |
| Verification | Question | Does the River Shannon originate from Dowra? |
| | Initial | Yes, **Dowra** is the source mountain of **River Shannon**. |
| | Generated | Yes, River Shannon starts from Dowra. |
| Paraphrase | Question | Who first ascended a mountain of Cathedral Peak (California)? |
| | Initial | **The person that first ascended** Cathedral Peak (California) is John Muir. |
| | Generated (en-de) | The first person to climb Cathedral Peak (California) is John Muir. |
| | Generated (en-ru) | The person who first climbed Mount Katty Peak (California) is John Muir. |

### *Input & Initial Verbalization*

Our framework requires at least one available verbalized answer per question to build upon it and extend it into multiple diverse paraphrased responses. Therefore, the generation workflow from [20] is adopted as a first step and used to generate the initial responses. This step's inputs are the questions, the SPARQL queries, and the hand-crafted natural language answer templates.

### *Entity-Type Identification though Named Entity Recognition*

Named Entity Recognition (NER) step recognizes and classifies named entities into predefined categories, for instance, persons, organizations, locations, etc. Our aim here is to identify the entity category (or entity-type) and span and replace it with a predefined value in the response. This stage allows us to accomplish more general verbalization since question entities are swapped with their type categories. The whole process is performed in 2 steps: 1) A pre-trained NER [18] model is employed to locate entities in the initial generated responses. Discovered entities are replaced with their type category such as "ORG, PRODUCT, LOC, PERSON, GPE". 2) A predefined dictionary is used to substitute the type categories with different words such as "the organization, the person, the country". Table 3 presents a generated example from the entity-type identification step.

### Gender Identification

In this step, we create new responses by replacing the question entities with their corresponding pronouns, e.g. "he, she, him, her". This is done by identifying the entity's gender. In particular, we query the KG with a predefined SPARQL query that extracts the gender of the given entity. Based on the position of the entity in the answer, we replace it with the appropriate pronoun. Table 3 illustrates a generated example from the gender identification step. In peer-review process, we verify the dataset to avoid any bias in the genders considering we extract gender information from DBpedia and sometime KG data quality is not perfect.

### New Verification Template

Considering that, on verification questions, all triple data is given (head, relation, tail). We introduce a verbalization template that interchanges the head and tail triple information and generate more diverse responses. Table 3 provides a generated example from this process.

### Paraphrase through Back-Translation

After having assembled sufficient answers for each question, we employ a paraphrasing strategy through a back-translation approach. In general, back-translation is when a translator (or team of translators) interprets or re-translate a document that was previously translated into another language back to the original language. In our case, the two translators are independent models, and the second model has no knowledge or contact with the original text.

In particular, inspired by [11,14], our initial responses alongside the new proposed answer templates are paraphrased using transformer-based models [39] as translators. The model is evaluated successfully on the WMT'18[8] dataset that includes translations between different languages. In our case, we perform back-translation with two different sets of languages: 1) Two transformer models are used to translate the responses between English and German language (en→de→en). 2) Another two models are used to translate between English and Russian language (en→ru→en). Here it is worth mentioning that we also forwarded output responses from one translation stack into the other (e.g., en→de→en→ru→en). In this way, we generate as many as possible different paraphrased responses. Please note that the selection of languages for back translation was done considering our inherited underlying model's accuracy in machine translation tasks on WMT'18. Table 3 illustrates some examples from our back-translation approach.

### Rectify Verbalization

After collecting multiple paraphrased versions of the initial responses, the last step is to rectify and rephrase them to sound more natural and fluent. The rectification step of our framework is done through a peer-review process to ensure the answers' grammatical correctness. Finally, by the end of this step, we will have at least two and at most eight diverse paraphrased responses per question, including the initial answer.

---

[8] http://www.statmt.org/wmt18/translation-task.html

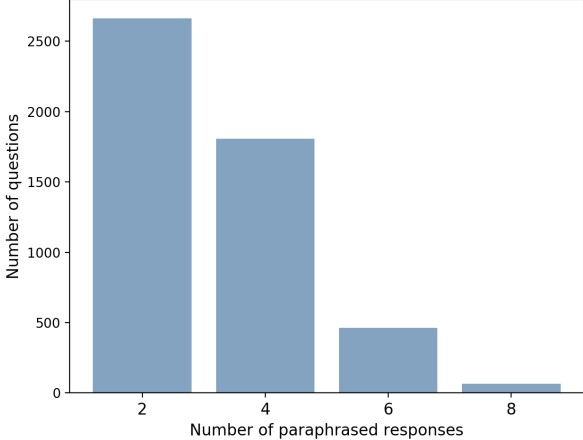

Fig. 2: Total paraphrased responses per question.

### 3.2 Dataset Statistics

We provide dataset insights regarding its total paraphrased results for each question and the percentage of generated answers from each module on our framework. Figure 2 illustrates the distribution of 5000 questions of ParaQA based on a total number of paraphrased responses per question. As seen from the figure, more than 2500 questions contain at most two paraphrased results. A bit less than 2000 questions include at most four answers, while around 500 have no less than six paraphrased answers. Finally, less than 100 examples contain at most eight paraphrased results. Figure 3 depicts the percentage of generated answers for each step from our generation workflow. The first step (input and initial verbalization) provides approximately 30% of our total results, while the next three steps (entity type identification, gender identification, and new verification templates) produce roughly 20% of responses. Finally, the back-translation module generates no less than 50% of the complete paraphrased answers in ParaQA.

## 4 Availability and Sustainability

**Availability** The dataset is available at a GitHub repository[9] under the Attribution 4.0 International (CC BY 4.0)[10] license. As a permanent URL, we also provide our dataset through figshare at https://figshare.com/projects/ParaQA/94010. The generation framework is also available at a GitHub repository[11] under the MIT License[12]. Please note, the dataset and the experiments reported

---

[9] https://github.com/barshana-banerjee/ParaQA
[10] https://creativecommons.org/licenses/by/4.0/
[11] https://github.com/barshana-banerjee/ParaQA_Experiments
[12] https://opensource.org/licenses/MIT

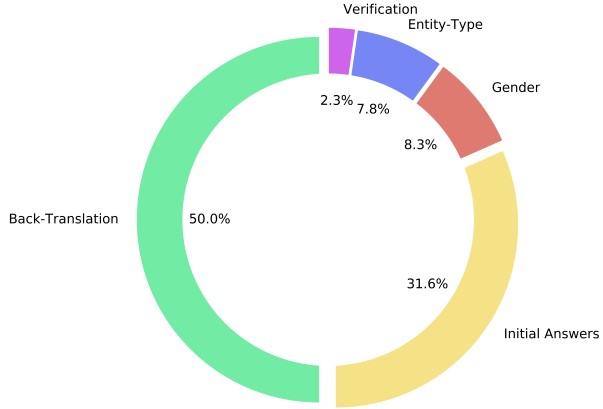

Fig. 3: Percentage of generated results from each step.

in the paper are in two different repositories due to the free distributed license agreement.

**Sustainability** The maintenance is ensured through the CLEOPATRA[13] project till 2022. After that, the maintenance of the resource will be handled by the question and answering team of the Smart Data Analytics (SDA)[14] research group at the University of Bonn and at Fraunhofer IAIS[15].

## 5   Experiments

To assure the quality of the dataset and the advantage of having multiple paraphrased responses, we perform experiments and provide baseline models, which researchers can use as a reference point for future research.

### 5.1   Experimental Setup

**Baseline Models**
For the baselines, we employ three sequence to sequence models. Sequence to sequence is a family of machine learning approaches used for language processing, and used often for natural language generation tasks. The first model consists of an RNN [24] based architecture, the second uses a convolutional network [15], while the third employs a transformer network [39] .

---

[13] http://cleopatra-project.eu/
[14] https://sda.tech/
[15] https://www.iais.fraunhofer.de/

### Evaluation Metrics

**BLEU (Bilingual Evaluation Understudy)** BLEU score introduced by [28] is so far the most popularly used machine translation metric to evaluate the quality of the model generated text compared to human translation. It aims to count the n-gram overlaps in the reference by taking the maximum count of each n-gram, and it clips the count of the n-grams in the candidate translation to the maximum count in the reference. Essentially, BLEU is a modified version of precision to compare a candidate with a reference. However, candidates with a shorter length than the reference tend to give a higher score, while the modified n-gram precision already penalizes longer candidates. Brevity penalty (BP) was introduced to rectify this issue and defined as:

$$BP = \begin{cases} 1, & c \geq r \\ exp(1 - \frac{r}{c}), & c < r \end{cases} \tag{1}$$

Where it gets the value of 1 if the candidate length $c$ is larger or equal to the reference length $r$. Otherwise, is set to $\exp(1 - r/c)$. Finally, a set of positive weights $\{w_1, ..., w_N\}$ is determined to compute the geometric mean of the modified n-gram precision. The BLEU score is calculated by:

$$BLEU = BP \cdot exp(\sum_{n=1}^{N} w_n log(P_n)), \tag{2}$$

where $N$ is the number of different n-grams. In our experiments, we employ $N = 4$ (which is a default value) and uniform weights $w_n = 1/N$.

**METEOR (Metric for Evaluation of Translation with Explicit ORdering)** METEOR score, introduced by [2], is a metric for the evaluation of machine-translation output. METEOR is based on the harmonic mean of unigram precision and recall, with recall weighted higher than precision.

BLEU score suffers from the issue that the BP value uses lengths that are averaged over the entire corpus level, leading to having individual sentences a hit. In contrast, METEOR modifies the precision at sentence or segment level, replacing them with a weighted F-score based on mapping uni-grams and a penalty function that solves the existing problem. Similar to BLEU, METEOR score can be in the range of 0.0 and 1.0, with 1.0 being the best score. Formally we define it as:

$$F_{mean} = \frac{P \cdot R}{\alpha \cdot P + (1 - \alpha) \cdot R},$$
$$Pen = \gamma \cdot \left(\frac{ch}{m}\right)^{\beta}, \tag{3}$$
$$METEOR = (1 - Pen) \cdot F_{mean}$$

where $P$ and $R$ are the uni-gram precision and recall respectively, and are used to compute the parametrized harmonic mean $F_{mean}$. $Pen$ is the penalty value

Table 4: BLEU score experiment results.

| Model | Input | One Response | Two Responses | Multiple Paraphrased |
|-------|-------|--------------|---------------|----------------------|
| RNN [24] | Question | 15.43 | 18.8 | 22.4 |
|          | SPARQL | 20.1 | 21.33 | 26.3 |
| Transformer [39] | Question | 18.3 | 21.2 | 23.6 |
|                  | SPARQL | 23.1 | 24.7 | 28.0 |
| Convolutional [15] | Question | 21.3 | 25.1 | **25.9** |
|                    | SPARQL | 26.02 | 28.4 | **31.8** |

and is calculated using the counts of chunks $ch$ and the matches $m$. $\alpha$, $\beta$ and $\gamma$ are free parameters used to calculate the final score. For our experiments we employ the common values of $\alpha = 0.9$, $\beta = 3.0$ and $\gamma = 0.5$.

***Training and Configurations***
The experiments are performed to test how easy it is for a standard sequence to sequence model to generate the verbalized response using as input only the question or the SPARQL query. Inspired by [20], during our experiments, we prefer to hide the query answer from the responses by replacing it with a general answer token. In this way, we simplify the model task to predict only the query answer's position in the final verbalized response.

Furthermore, we perform experiments with three different dataset settings. We intend to illustrate the advantage of having multiple paraphrased responses compared to one or even two. Therefore, we run individual experiments by using one response, two responses, and finally, multiple paraphrased responses per question. To conduct the experiments for the last two settings, we forward the responses associated with their question into our model. We calculate the scores for each generated response by comparing them with all the existing references. For the sake of simplicity, and as done by [16], the final score is the maximum value achieved for each generated response.

For fair comparison across the models, we employ similar hyperparameters for all. We utilize an embeddings dimension of 512, and all models consist of 2 layers. We apply dropout with probability 0.1. We use a batch size of 128, and we train for 50 epochs. Across all experiments, we use Adam optimizer and cross-entropy as a loss function. To facilitate reproducibility and reuse, our baseline implementations and results are publicly available[16].

### 5.2   Results

Table 4 and Table 5 illustrate the experiment results for BLEU and METEOR scores, respectively. For both metrics, the convolutional model performs the best.

---

[16] https://github.com/barshana-banerjee/ParaQA_Experiments

Table 5: METEOR score experiment results.

| Model | Input | One Response | Two Responses | Multiple Paraphrased |
|---|---|---|---|---|
| RNN [24] | Question | 53.1 | 56.2 | 58.4 |
| | SPARQL | 57.0 | 59.3 | 61.8 |
| Transformer [39] | Question | 56.8 | 58.8 | 59.6 |
| | SPARQL | 60.1 | 63.0 | 63.7 |
| Convolutional [15] | Question | 57.5 | 58.4 | **60.8** |
| | SPARQL | 64.3 | 65.1 | **65.8** |

It outperforms the RNN and transformer models in different inputs and responses. Here, it is more interesting to notice that all models perform better with multiple paraphrased answers than one or two responses. At the same time, the scores with two answers are better than those with a single response. Hence, we can assume that the more paraphrased responses we have, the better the model performance. Concerning the experiment inputs (Question, SPARQL), as indicated by both metrics, we obtain improved results with SPARQL on all models and responses. As expected, this is due to the constant input pattern templates that the SPARQL queries have. While with questions, we end up having a different reworded version for the same template. We expect the research community to use these models as baselines to develop more advanced approaches targeting either single-turn conversations for QA or answer verbalization.

## 6   Reusability and Impact

ParaQA dataset can fit in different research areas. Undoubtedly, the most suitable one is in the single-turn conversational question answering over KGs for supporting a more expressive QA experience. The dataset offers the opportunity to build end-to-end machine learning frameworks to handle both tasks of query construction and natural language response generation. Simultaneously, the dataset remains useful for any QA sub-task, such as entity/relation recognition, linking, and disambiguation.

Besides the QA research area, the dataset is also suitable for Natural Language Generation (NLG) tasks. As we accomplish in our experiments, using as input the question or the query, the NLG task will generate the best possible response. We also find ParaQA suitable for the NLP area of paraphrasing. Since we provide more than one paraphrased example for each answer, researchers can experiment with the dataset for building paraphrasing systems for short texts. Furthermore, our dataset can also be used for the research involving SPARQL verbalization, which has been a long-studied topic in the Semantic Web community [26,27,13].

## 7   Conclusion and Future Work

We introduce ParaQA – the first single-turn conversational question answering dataset with multiple paraphrased responses. Alongside the dataset, we provide a semi-automated framework for generating various paraphrase responses using back-translation techniques. Finally, we also share a set of evaluation baselines and illustrate the advantage of multiple paraphrased answers through commonly used metrics such as BLEU and METEOR. The dataset offers a worthwhile contribution to the community, providing the foundation for numerous research lines in the single-turn conversational QA domain and others. As part of future work, we look to work on improving and expanding ParaQA. For instance, supporting multi-turn conversations together with paraphrased questions is also in our future work scope.

## Acknowledgments

The project leading to this publication has received funding from the European Union's Horizon 2020 research and innovation program under the Marie Skłodowska-Curie grant agreement No. 812997 (Cleopatra).

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
