# OpenReview forum: "ParaQA: A Question Answering Dataset with Paraphrase Responses for Single-Turn Conversation"
_eswc-conferences.org/ESWC/2021/Conference/Resources_Track — ESWC 2021 Resources_

### Official Review · AnonReviewer4 · 2021-01-11
**The authors presents a question answering approach based on multiple paraphrased responses for conversations over knowledge graphs.**

**Rating:** 2
**Confidence:** 2

**Review:**

The paper argues that answer paraphrasing improves the machine learning models’ performance for single-turn conversations. The authors proposed ParaQA as a question answering dataset with multiple paraphrase responses for single-turn conversation over KGs.

**Anonymity:**

Yes, I would like my review to remain anonymous.

**Strong Points:**

The contributions of the paper seems well structured with clear articulation of related work. This is not my core research area, but the automation framework for generating multiple paraphrase responses for each question sounds quite interesting. There is also some evaluation baseline.

**Subreviewer:**

I submitted this review.

**Weak Points:**

While the level of evaluation presented is commendable. Albeit for generalisation, a discussion of the practicalities of the approach particularly the effort associated with semi-automated multiple paraphrase responses for questions would suffice.

---

> ### Author Rebuttal · Authors · 2021-01-27
>
> We thank the reviewer for carefully reviewing our submission and for the thoughtful comments and feedback. We promise to address the weak point on the final version of the paper.

---

### Official Review · AnonReviewer1 · 2021-01-14
**A valuable resource for KBQA improvements**

**Rating:** 2
**Confidence:** 5

**Review:**

This paper presents a new dataset for KBQA (knowledge base question answering). The unique characteristic of this dataset, ParaQA, is that each question corresponds to multiple (2 to 8) paraphrased answers, whereas all existing datasets have only one answer for each question. This one-to-many relationship allows a KBQA model to be more robust and generalisable. The dataset and the associated tools have been made available on GitHub.

Evaluation of three QA models shows that the increase in the number of answers improves model performance, in terms of BLEU and METEOR.

**Strengths**:

+ An interesting and novel take on KBQA improvement. Specifically, instead of changing model architecture, the authors proposed a dataset with a new twist (multiple answers).
+ The paper is well-motivated, well-written and easy to follow.
+ The dataset and tools have been made available on GitHub.

**Weaknesses**:

- Most of the existing KBQA datasets don't have verbalised answers. Thus, it remains a question how useful a model trained/optimised on this dataset on existing datasets.


**Detailed comments*:

* I find the focus on "single-turn conversation" unnecessary. Sure, you can call such a task as single-turn conversations. However, it's just a plain QA task of a single question and answer.

* Which BLEU is it? BLUE-1, 2, 3, 4, or their average?

* For KBQA, the usual performance metrics are F1, accuracy or EM (exact match), but not these generation-based metrics. A discussion on the implication of this difference would be useful.

* On page 12, you said "In our opinion, a question can be regarded as correctly answered if its formal query has been built correctly.". What do you mean by "forward"?

* The experiments use three models that are based on RNN, Transformer and CNN. However, none of these models is specifically designed for the KBQA task. I wonder whether the proposed dataset would induce similar performance gains on recent, strong models.

## Rebuttal
Your response addresses some of my questions. In your revision, please do clarify the few issues I raised in the review: the purpose of this dataset, the rationale of the metrics and the way the dataset is constructed ("forward").


**Anonymity:**

Yes, I would like my review to remain anonymous.

**Strong Points:**

See **Strengths** above.

**Subreviewer:**

I submitted this review.

**Weak Points:**

See **Weaknesses** above.

---

> ### Author Rebuttal · Authors · 2021-01-27
>
> We thank the reviewer for carefully reviewing our submission and for the thoughtful comments and feedback. We would like to address these and provide further details on four main points:
>
> 1) Most of the existing KBQA datasets don't have verbalised answers. Thus, it remains a question how useful a model trained/optimised on this dataset on existing datasets.
>
>     Answer: The dataset we provide addresses the research field of KBQA. Since the paper's main focus is on multiple (paraphrased) answers, the experiments were explicitly done to target the task of answer generation. Therefore we do not cover the complete KBQA task on the experiments. So we provided baseline models only for the answer generation task. We will further clarify this in the next version of the paper.
>
>
> 2) Which BLEU is it? BLUE-1, 2, 3, 4, or their average?
>
>      Answer: We employ a default N=4 number of different n-grams (BLUE-4) with uniform weights (1/N).
>
>
> 3) Regarding the metrics used for our experiments.
>
>      Answer: Since our experiments focus only on answer verbalization, we stick with metrics such as BLEU and METEOR. In case of a complete KBQA task then we had to also report F1 and others.
>
>
> 4) On page 12, what do you mean by "forward"?
>
>     Answer: Our experiments focus on whether multiple paraphrased answers help on obtaining better results compared to settings where we have only 1 or 2 answers. In our case, having multiple paraphrased answers means that the model will have to learn from multiple gold examples. Based on the number of answers we have, we create that many inputs with the same question. The word “forward” is used to describe the process of providing each input (question+answer) to the model and extracting the result. We will further clarify this in the paper.

---

> > ### Comment · AnonReviewer1 · 2021-02-01
> > **Thanks for your rebuttal**
> >
> > Your response addresses some of my questions. In your revision, please do clarify the few issues I raised in the review: the purpose of this dataset, the rationale of the metrics and the way the dataset is constructed ("forward").

---

### Official Review · AnonReviewer3 · 2021-01-14
**In this paper the question-answering dataset, ParaQA, is presented, where for each question a set of maximum eight paraphrased answers is provided. The questions from LC-QuAD dataset are used along with the respective verbalized answers from VQuAnDA (which is created from the same research group). The semi-automated framework for generating these paraphrased answers is described.**

**Rating:** 1
**Confidence:** 4

**Review:**

This is a paper very well written and easy to read. As this work generates a QA dataset with paraphrased answers it is within the scope of this conference, especially it is related to the subtracks KGs, and NLP and Information retrieval. To the best of my knowledge there is no other QA dataset over KGs with paraphrased answers. Hence the novelty of this work is evident. However, the motivation and impact of this research work is not very clear. The authors argue that “answer paraphrasing improves the machine learning models’ performance for single -turn conversations (involving question answering over KGs)”, however to illustrate this they use some sequence-to-sequence translating tools and not QA systems. So, it is left to the reader to understand how the QA over KGs will be improved with these paraphrased responses. I believe that the usability of this work within this domain would be better illustrated if they used for the evaluation some QA system instead. In Section 6 they argue also that this dataset can be used for research in SPARQL verbalization which is also quite unclear. Finally, it is very important to note that there is a significant amount of research on paraphrasing (e.g. [1,2]) that is not mentioned in the related work section of this paper. I believe that authors should also evaluate their framework against other existing paraphrasing tools like the aforementioned ones.

[1] A. B. Siddique, Samet Oymak, and Vagelis Hristidis. 2020. Unsupervised Paraphrasing via Deep Reinforcement Learning. In Proceedings of the 26th ACM SIGKDD International Conference on Knowledge Discovery & Data Mining (KDD '20). Association for Computing Machinery, New York, NY, USA, 1800–1809. DOI:https://doi.org/10.1145/3394486.3403231

[2] Ankush Gupta, Arvind Agarwal, Prawaan Singh, and Piyush Rai. 2018. A deep generative framework for paraphrase generation. In Thirty-Second AAAI Conference on Artificial Intelligence

**Anonymity:**

Yes, I would like my review to remain anonymous.

**Strong Points:**

-Well written paper, relevant to the conference

-Novel work

-Publicly available

-Good evaluation results

**Subreviewer:**

I delegated this review to a subreviewer.

**Weak Points:**

-Related work on paraphrasing techniques is missing

-It is not tested against other paraphrasing tools

-The motivation is quite unclear

---

> ### Author Rebuttal · Authors · 2021-01-27
>
> We thank the reviewer for carefully reviewing our submission and for the thoughtful comments and feedback. We would like to address these and provide further details on three main points:
>
> 1) Related work on paraphrasing techniques is missing
>
>      Answer: Currently, our related work focuses on existing datasets and different approaches on how to create such datasets. Since paraphrasing is only one part of our construction framework we did not emphasize on it for the related work. We thank the reviewer for pointing this out, we will extend the related work with existing paraphrasing approaches on the final version of the paper.
>
>
> 2) It is not tested against other paraphrasing tools
>
>     Answer: In our case, paraphrasing is one of the steps used in the dataset construction framework. Even though we have investigated different existing paraphrasing techniques, we only applied one since the results from the paraphrasing module had to go through the peer-review process. We agree that it would be interesting to test with different paraphrasing tools and see whether the results are better. Most probably this can be a future work of our method.
>
>
> 3) The motivation is quite unclear
>
>      Answer: Our main motivation is to provide the first question answering dataset with multiple verbalized answers. At the same time, we want to show that having multiple verbalized answers affects positively on the results of the employed machine learning model. Hence, in our experiments, where we train baseline models only for the verbalization task and we indicate that using multiple verbalized answers the same model produces higher results compared to when we have 1 or 2 answers.
> We will further clarify the motivation of our work in the introduction of the paper.

---

### Official Review · AnonReviewer2 · 2021-01-15
**Not really sure about the contribution level**

**Rating:** 1
**Confidence:** 4

**Review:**

The authors present a new dataset for QA over KGraphs focused on having multiple verbalizations of the answers in order to improve the naturality of the interaction with the user in single-turn conversation.

The dataset is interesting, however, my main concern is that I am not sure if it's novel enough so as to be published as a dataset itself or an extension of the previous ones already published by the authors (in two previous publications):

- I fail to see a huge difference between regular Question Answering task and Single-turn conversation QA. In both of them a user inputs a query with all the information in it (no memory, no anaphora of any other element in the conversation, no further interaction) ... to the point that the authors directly take their previous datasets on QA over KGraphs in order to have the starting point (one query verbalization + SPARQL output + one answer verbalization). While the lack of paraphrasing is true in QA datasets (as the main goal is to have just an answer, not a "good-looking" one),  the "only" (big quotes here) difference would be to try to verbalize the answer in a more natural way, but this would not only fall into answer paraphrasing, but closer to knowledge graph verbalization (at least with factual queries, not with complex ones - where aggregations/operators might appear) and general paraphrasing task. Somehow, I find the paper more an extension of VQuAnDA rather than a completely new dataset. I miss some paraphrasing as well for the input queries in order to further train a conversational agent to deal with different shapes of the same query. Regarding Section 6 and reusability, some of the claims do not come from this particular contribution, but from the datasets it builds upon.

- Regarding the approach itself:
	* How queries with multiple entities in them are handled?
	* In the automatic back-translation paraphrasing, there might be a lot of information lost in the translation. While this is peer-reviewed afterwards, the details of such peer-review are not clear.
	* The evaluation is not very fair, as giving the maximum values only does not account for the potential noise that several paraphrases might have introduced. It would be interesting to have the average values for each paraphrases to have a little bit more information about the procedure. Is not surprising that the evaluation values are better, more options are considered and we get always the best of them (without taking into account that there might be very bad ones - which should be already filtered out in the peer-review, but we need insight even though).


Minor comment:
- Table 2 => in the answer verbalizations there are some examples that are not very natural (e.g., "The nationality of Lawrence ... is Great Britain" seems a little bit artificial in fact).
- What are the kind of answers that are more prone to be paraphrased?
- Figure 3 is not very readable.

Typos:
- page 5: judges (in the paraQA example) => judge
- page 5: As can be seen => As it can be seen
- page 11: R' in eq 3


Strong points




**Anonymity:**

Yes, I would like my review to remain anonymous.

**Strong Points:**

- Built on top of previous good datasets
- Availability and sustainability
- Interesting approach to extend VQuAnDA with paraphrasis

**Subreviewer:**

I submitted this review.

**Weak Points:**

- Possible lack of novelty, it seems more an extension of datasets already published by the authors (with the corresponding papers)

---

> ### Author Rebuttal · Authors · 2021-01-27
>
> We thank the reviewer for carefully reviewing our submission and for the thoughtful comments and feedback. We would like to address these and provide further details on four main points:
>
> 1) How queries with multiple entities in them are handled?
>
>     Answer: Regarding the dataset generation process, we take care during the peer-review checking that all entities are mentioned correctly on both questions and verbalised answers.
> For the experiments, where we only generate the verbalised answer, we do not have any exception for multiple entities. The model should be able to position the question entities on the generated answer correctly. We only introduce an answer token for the answer entities since they do not appear on the question, and therefore the model would fail on predicting them.
>
>
> 2) In the automatic back-translation paraphrasing, there might be a lot of information lost in the translation. While this is peer-reviewed afterwards, the details of such peer-review are not clear.
>
>     Answer: Yes, we have the peer-review process guidelines that affect the results of the complete dataset construction framework and not only the paraphrasing step. We will include them on the final version of the paper, thank you for pointing this out.
>
>
> 3) Regarding the evaluation by selecting the maximum value.
>
>     Answer: As we mention, our experiments also aim to prove that multiple answer verbalizations are more useful for machine learning models compared to having 1 or 2 answers. Here by “useful” we mean that the model provides better results with them. We employ simple baseline models for the experiments. Using the maximum value for our experimental settings are adopted from Gupta et al. (2019), which we mention and cite in the paper.
>
>
> 4) Possible lack of novelty, it seems more an extension of datasets already published by the authors (with the corresponding papers):
>
>      Answer: We would like to argue with this point since we consider the following three points as our novelty:
>      1) We present the first QA dataset with multiple paraphrased answers. No other dataset before had multiple answers and it is something that the KGQA field is lacking.
>      2) We provide a detailed stepwise framework on how to create such a dataset.
>      3) We train evaluation baseline models to determine our dataset’s quality and also to prove the effectiveness of having multiple paraphrased answers.
>
>   Furthermore, the rationale behind using the existing datasets is:
>       * Our focus is not on question generation, and we focus on answer formulation. This is one key element lacking in the existing datasets. Hence, rather than reinventing the wheel to create new questions, we supplement existing datasets with a wide range of paraphrase answers for wider community adaptation.
>
> Finally, we would like to thank the reviewer for the reported minor comments and typos; we will address and fix them on the final version.

---

> > ### Comment · AnonReviewer2 · 2021-02-01
> > **Not really sure about the contribution level - Rebuttal**
> >
> > After reading the answers to my concerns, I'm moving my evaluation to Borderline/weak accept.
> > However, I'd encourage authors to improve the evaluations regarding other paraphrasis datasets / tools as suggested by another reviewer (in the end, that's what the dataset is about). Besides, regarding answer 3, I'd also strongly suggest to improve the evaluation information adding the mean values and the std across all the answer verbalizations. The fact that that method was already used in another publication does not make it a global fair comparison: if you add up to 8 verbalizations, you might be including a lot of noise which maximum values would filter (it's about giving more valuable information in the paper itself).

---

### Decision · Program_Chairs · 2021-02-23

**Decision:**

Accept

**Comment:**

The dataset contains 5000 question-response pairs with between two and eight paraphrased responses for single-turn QA over knowledge graphs. The reviewers positively highlight the quality of writing, availability, sustainability, and evaluation, while they raise issues regarding the novelty (also as compared to two previous papers), missing related work on paraphrasing techniques, and a lack of against other paraphrasing tools. Ultimately, all reviewers suggest accepting the paper.